# The Association between Delivery during the COVID-19 Pandemic and Immediate Postpartum Maternal Cognitive Function

**DOI:** 10.3390/jcm9113727

**Published:** 2020-11-20

**Authors:** Hagai Hamami, Eyal Sheiner, Tamar Wainstock, Elad Mazor, Talya Lanxner Battat, Asnat Walfisch, Tamar Kosef, Gali Pariente

**Affiliations:** 1Faculty of Health Sciences, The Joyce and Irving Goldman Medical School, Ben-Gurion University of the Negev, Beer-Sheva 8410501, Israel; HagaiHamami@gmail.com; 2Department of Obstetrics and Gynecology, Soroka University Medical Center, Ben-Gurion University of the Negev, Beer-Sheva 8410501, Israel; EladMa@clalit.org.il (E.M.); talyalan@gmail.com (T.L.B.); galipa@bgu.ac.il (G.P.); 3Department of Public Health, Faculty of Health Sciences, Ben-Gurion University of the Negev, Beer-Sheva 8410501, Israel; wainstoc@bgu.ac.il; 4Department of Obstetrics and Gynecology, Hadassah Hebrew University Medical Center, Mount Scopus, Jerusalem 9112001, Israel; asnatwalfisch@yahoo.com; 5Department of Psychiatry, Soroka University Medical Center, Ben-Gurion University of the Negev, Beer-Sheva 8410501, Israel; tamar.kosef@gmail.com

**Keywords:** COVID-19, cognitive function, symbol digit modalities test (SDMT), attention function index (AFI)

## Abstract

Survivors of the 2003 SARS epidemic were found to have higher rates of adverse mental conditions. This study aimed to assess cognitive function in women delivering during the COVID-19 pandemic, as compared to women who delivered before the COVID-19 pandemic. A cohort study was performed during the immediate postpartum period of women delivering singletons at term. Cognitive function was assessed using an objective neurocognitive test (Symbol Digit Modalities Test SDMT90, SDMT4) and a subjective self-estimation questionnaire (Attention Function Index AFI). The exposed group was recruited during the COVID-19 outbreak in Israel (May 2020), whereas the comparison group consisted of women delivering at the same medical center before the COVID-19 pandemic (2016–2017). Multivariable regression models were constructed to control potential confounders. There were 79 parturients recruited during the COVID-19 pandemic and compared with 123 women who delivered before the COVID-19 pandemic. Women delivering during the COVID-19 pandemic scored lower in the subjective AFI test compared to the unexposed group (70.0 ± 15.4 vs. 75.1 ± 14.7, *p* = 0.018). However, no significant difference was found in the objective SDMT tests scores. These results remained similar in the multivariable regression models when controlling for maternal age, ethnicity and time from admission to assessment, for AFI, SDMT90 and SDMT4 scores (*p* = 0.014; *p* = 0.734; *p* = 0.786; respectively). While no significant difference was found in objective tests, our findings propose that the exposure to the COVID-19 pandemic is independently associated with a significant decrease in subjective maternal cognitive function during the immediate postpartum period.

## 1. Introduction

Coronavirus disease 2019 (COVID-19), was first identified in December 2019 in China’s Hubei province, and has since become a global concern, resulting in the ongoing coronavirus pandemic [1]. The World Health Organization (WHO) declared the outbreak a Public Health Emergency of International Concern on 30 January, and a pandemic on 11 March. As of 29 September 2020, more than 33 million cases of COVID-19 have been reported in more than 188 countries and territories, resulting in more than 1,000,000 deaths [2].

A sudden outbreak of a disease poses threat to the mental health of affected people. Indeed, depression, anxiety and other negative psychological effects have all been found among the survivors of the 2003 SARS epidemic [3].

Conflicting reports have been published regarding mental health problems due to outbreak of COVID-19 and mass quarantine. A recent study assessed the mental health status of Chinese people from Hubei province during the COVID-19 pandemic and demonstrated higher rates of anxiety, depression, and alcohol consumption [4]. The study also suggested that people aged 21–40 years old were more psychologically vulnerable during the pandemic [4]. One particular group that might be negatively affected by the pandemic is pregnant women and women during the postpartum period, since stress is also thought to have a significant effect on pregnancy outcomes [5]. Nevertheless, another cohort study that assessed risk for depression among pregnant women hospitalized in a high-risk pregnancy ward during the COVID-19 pandemic demonstrated comparable risk for depression compared to pregnant women hospitalized before the pandemic [6].

Generally, pregnancy is known to affect the maternal emotional state, with depression being a common complication of pregnancy and the postpartum period [7,8]. Although postpartum depression is a common condition that may affect both the mother and the offspring [9], along with other associated maternal mental health problems, it could develop unnoticed if not thoroughly screened for [10].

Postpartum depression occurs in up to 19% of all pregnancies, while the first six months after delivery are thought to bear increased risk for the onset of depression [11]. As per DSM-V definition, this condition occurs up to four weeks after delivery, yet many experts in the field define postpartum depression as occurring anytime within the first year postpartum, irrespective of the time of onset [12]. During pregnancy, the third trimester is when women face the greatest risk of developing depression [13]. Pregnancy may affect maternal cognitive function, mainly in the fields of memory and attention, with some studies suggesting both short-term and long-term cognitive deficits during pregnancy and postpartum [14]. Notably, not all studies have found a significant influence of pregnancy on cognitive function.

Moreover, since subjective reports show that pregnant women tend to rate their own memory as being negatively affected by pregnancy [15], it remains questionable whether this cognitive decline is the result of neural and physiological pregnancy-related changes, or the result of one’s own subjective perception of cognitive function and mental state [14]. Studies have described a complex relationship between depression and a decline in cognitive function, especially in the elderly population, as one condition could be seen as a risk factor or a complication of the other, and vice versa [16,17].

In an effort to understand the influence of delivering during the COVID-19 pandemic on maternal mental health, we aim to assess subjective and objective cognitive function in women delivering during the COVID-19 pandemic compared to women who delivered before the COVID-19 pandemic.

## 2. Materials and Methods

### 2.1. Population and Settings

The study population was comprised of two groups. The unexposed group derives from published data that included 123 women who were enrolled before the COVID-19 pandemic had occurred [15]. During the pandemic 112 women were approached, of whom 17 refused to participate in the study (mainly due to privacy concerns), and 16 did not meet the inclusion criteria. The remaining 79 women were included in the final analysis and constituted the exposed group. Women who delivered a healthy singleton at term (≥37^0/7^ gestational weeks), without complications, were included. In order to evaluate the independent association between exposure to the COVID-19 pandemic and cognitive function, any delivery necessitating intervention was excluded. Such interventions included: repair of a third or fourth degree perineal tear, revision (inspection with/without removal of content or repair) of the uterine cavity or birth canal under anesthesia (general or epidural), and blood product transfusion. Illiterate women and women who did not provide an oral and written consent to participate in the study were also excluded.

The study was approved by the local Institutional Review Board (IRB approval # 0079-20-SOR).

### 2.2. Study Design

A cohort study was performed in women during the immediate postpartum period (1–3 days postpartum), focusing on healthy women hospitalized in the maternity wards of the Soroka University Medical Center (SUMC). Enrollment and data collection were performed during May 2020. Each woman was evaluated at a single time point.

The exact assessment of cognitive function requires both time and expertise. As a result, validated questionnaires that serve as screening tests are widely used in similar studies; concretely, the Symbol Digit Modality Test and the Attention Function Index are used to assess objective and subjective cognitive function, respectively [18,19].

The research team approached parturients hospitalized in the maternity wards (1–3 days postpartum) and offered them participation following an oral and written explanation regarding the study’s course and purpose. Eligible women were asked to sign the consent form and fill out the following questionnaires.

The Symbol Digit Modalities Test (SDMT) is a neurocognitive test, designed to assess the decline in cognitive functions of tested individuals, focusing on decrease in motor speed, visual scanning and concentration [18]. While filling the SDMT, subjects are limited to either 90 s or 4 min (since both time frames have been described in the literature, both were used in the current study and designated SDMT90 and SDMT4, respectively). In this test, the sum of the correct answers is a continuous variable (range between 0 to 225). The correlation between the two test’s variations was also analyzed.

The SDMT is an accepted tool implemented in a wide range of research areas and clinical settings for the assessment of patients of different ages and conditions, including children and adults with multiple sclerosis, elderly patients with Alzheimer’s disease, and breast cancer patients [20,21,22].

The Attention Function Index (AFI) is a subjective test, which measures the tested individual’s self-estimation of one’s own cognitive function, focusing on working memory and concentration [19]. The subjects are asked to mark their degree of agreement (on a scale of 0 to 100) with each of the 13 phrases included in the AFI test.

The Edinburgh Postnatal Depression Scale (EPDS) was developed and published by Cox et al., to improve the estimation of risk for depression among pregnant and postpartum women [23]. As opposed to the DSM-5 definition for postpartum depression, which was constructed to formally diagnose major depression in postpartum women within a defined time frame, the EPDS test serves as an initial screening tool designed to demonstrate the likelihood of perinatal depression and is widely used based on the recommendations of the American College of Obstetricians and Gynecologists [24,25]. It is a questionnaire that consists of 10 self-estimation phrases, in which women are asked to rate their mood in the past seven days, choosing between 4 possible answers for each phrase or question. Women are then classified according to their EPDS score: a score of less than 10 defines a woman at low risk for depression, while a score of 10 or greater defines women at high risk for depression.

Each woman from both the exposed and the comparison group answered questions regarding her socioeconomic state, obstetrical history, and current pregnancy course.

### 2.3. Statistical Analysis

Dependent variables included the SDMT, AFI and EPDS scores while the main independent variable was the exposure to the COVID-19 pandemic (i.e., enrollment during the COVID-19 pandemic and its outbreak in Israel). Background variables assessed included maternal demographic and pregnancy data, as well as delivery course and immediate outcome. Statistical analysis was performed using SPSS version 23.0. Comparison of continuous variables was performed using Student’s t–test or Mann–Whitney U-test according to the pattern of distribution (normal or not, respectively). Chi-square test was used to examine differences in the distribution of categorical variables.

A multivariable linear regression model was constructed to examine the relationship between the independent and dependent variables, while adjusting for potential confounding factors as ascertained from the univariate analysis. Variables assessed as potential confounders included ones known to be associated with maternal cognitive function or found to be significant in the univariate analysis. These included maternal age, ethnicity and time from admission to assessment.

Sample size was calculated on the basis of the following assumptions: The power of the study was defined as 80%, bilateral hypothesis with statistical significance set at 95% (α error of 5%). A sample of 79 women was calculated to be sufficient to detect a difference of 3 points in the average score of the SDMT test.

## 3. Results

### 3.1. Study Groups

A total of 79 women who delivered during the COVID-19 pandemic and the COVID-19 outbreak in Israel, and 123 women who delivered before the COVID-19 pandemic, were included in the final analysis.

Table 1 summarizes the demographic characteristics of both groups. Mean maternal age was comparable between the study group and the comparison group (28.1 ± 5.6 vs. 28. 3 ± 5.1 years, respectively, *p* = 0.836), as were country of birth, family status, background health, and ethnicity and nulliparity rates. According to the EPDS score, the exposed group had a slightly lower rate of high-risk for depression, yet this difference was not found to be statistically significant. A significant difference was found in the time passed between patient admission and assessment (1.0 ± 0.8 vs. 1.6 ± 1.2, days, respectively, *p* < 0.001).

### 3.2. Pregnancy and Delivery Characteristics

Pregnancy and delivery outcomes of both groups are presented in Table 2. No significant differences were noted between the groups in pregnancy course and outcome, including pregnancy screening tests (nuchal translucency, serum biomarkers, alpha-fetoprotein test and early and late ultrasound anatomical survey), complication rates (gestational diabetes and preeclampsia) and delivery characteristics. All newborns had a normal 5 min Apgar score.

### 3.3. Cognitive Tests

Table 3 summarizes maternal objective cognitive function results, as assessed by the SDMT90 and SDMT4 tests, alongside results of the subjective AFI test, representing maternal self-estimation of cognitive function. Women who gave birth during the COVID-19 pandemic scored significantly lower in the subjective AFI test. Mean AFI score was 70.0 ± 15.4 in the exposed group, while the unexposed group mean AFI score was 75.1 ± 14.7 (*p* = 0.018). However, no difference was seen between the groups’ objective tests scores.

### 3.4. Multivariable Analyses

Table 4 summarizes the multivariable regression model for maternal subjective cognitive function, controlling for maternal age, ethnicity and time passed from admission to assessment. Delivery during the COVID-19 pandemic was found to be independently associated with the subjective decline in cognitive function (AFI score, Beta = −5.6; 95% CI: −10.09 to −1.13, *p* = 0.014). However, as seen in Table 5 and Table 6, the objective cognitive function tests (SDMT90 and SDMT4) were not independently associated with delivery during the COVID-19 pandemic, whereas controlling for maternal age, ethnicity, and length of time passed from admission to assessment.

## 4. Discussion

Our study found that delivering during the COVID-19 pandemic is independently associated with a significant decrease in maternal subjective cognitive function during the immediate postpartum period. Nevertheless, no significant difference was found in objective tests.

Several studies have evaluated effects of natural disasters or world pandemics on offspring development [26,27]. Project Ice Storm, which refers to a series of prospective studies on women who were pregnant during one of Canada’s worst natural disasters in history, the January 1998 Quebec ice storm, was designed to study the effects of in utero exposure to varying levels of prenatal maternal stress (PNMS), resulting from an independent stressor on the children’s development from birth through childhood [28,29,30,31]. Jones et al. showed that higher levels of objective PNMS were associated with more externalizing problems (e.g., aggressive behavior), which was in part mediated by measurable differences in amygdala development in the offspring. This effect was shown to be stronger when the stress exposure occurred during more advanced gestational weeks [26]. Moreover, Cao-Lei et al. demonstrated how maternal cognitive appraisal of this natural disaster affected DNA methylation in their children 13 years after the ice storm, suggesting that maternal subjective perception of a natural disaster experienced during pregnancy has a widespread effect on offspring epigenetics [27].

A study by Nomura et al. evaluated the influence of in utero exposure to both maternal risk for depression and Hurricane Sandy on infant temperament [32]. When maternal risk for depression was assessed using the EPDS, their analysis showed interaction effects between prenatal maternal depression and Hurricane Sandy exposure, where prenatal maternal depression was associated with greater levels in activity, distress, approach, and shorter duration of attention of infants only when they were also exposed to Hurricane Sandy in utero [32]. These findings may suggest that maternal depression could bear additional negative effects on offspring development in the context of in utero exposure to a natural disaster. Therefore, we deduce that maternal decline in either emotional or cognitive function, when experienced during a natural disaster or a large-scale crisis, might be associated with more grave outcomes on maternal health and offspring development.

A study conducted at the SUMC discussed the association between maternal postpartum depressive state and maternal cognitive function during the immediate postpartum period [15]. The authors failed to find an association between maternal depressive state and maternal objective decline in cognitive function during the immediate postpartum period, but did find an association between maternal depressive state and maternal own subjective perception of cognitive decline, suggesting that the effect the maternal emotional state has on cognitive function might be purely subjective [15].

Zlatar et al. stated that subjective cognitive complaints are more likely related to symptoms of depression rather than concurrent cognitive impairment in a large cross-section of community-dwelling adults without a formal diagnosis of dementia [33]. While our study found subjective cognitive decline to be significantly different between women delivering during and before the COVID-19 pandemic, no difference between the two groups was seen regarding the risk for depression. Nevertheless, being a non-diagnostic tool, the possibility that the EPDS fails to distinguish between the true prevalence of maternal depression in both groups cannot be ruled out. Even if there is no difference in the true prevalence of maternal depression in both groups, when considering possible interaction effects such as those described by Nomura et al., the apparent decline in subjective cognitive function could be seen as an aggravation of maternal depression consequences that results from its coincidence with a large-scale crisis [32].

Crawley et al. reported that while pregnant women do not exhibit more cognitive difficulties compared with non-pregnant women, the former tend to rate themselves lower in terms of cognition, possibly stemming from a depressed mood, or due to misattributions based on cultural stereotypes of cognitive impairment during pregnancy [34]. This alone does not explain the apparent difference in our data since both the exposed group and the comparison group consisted of women in a similar postpartum period.

There are several limitations to our study. First, technical settings such as those affected by the surrounding environment (e.g., presence of the newborn or visitors; time of the day), may have affected the participants’ performance in the tests. In addition, the SDMT, much like other objective tests, may not correctly detect mild cognitive impairment, partly owing to the examined individual ability to make a conscious effort during short periods of testing. A selection bias is also possible, since women who suffer from an objective or subjective decline in cognitive function might have refrained from participating in our study, possibly due to lack of anonymity during data collection. Furthermore, we had no access or knowledge regarding maternal history of cognitive abilities, which may have impacted our results as well, since even depression, anxiety and stress are possible confounders. Additionally, this study measured each mother’s cognition at a single time-point, and thus cannot identify a longitudinal pattern of objective and subjective cognitive changes.

There is a growing literature that suggests it is important to assess both categorical and dimensional aspects of mental disorders, particularly as there may be individuals who display symptoms of disorder (including cognitive impairment) and experience significant distress but whose symptoms do not meet diagnostic criteria. Moreover, as cognitive function is complex and encompasses many aspects of mental abilities and intellectual functions (e.g., judgment, working memory, attention and decision making), it cannot be properly evaluated using a single test such as the SDMT, since it only challenges a limited set of abilities and functions. The SDMT and the AFI constitute one objective and one subjective assessment tools respectfully, while the use of more subjective and objective tools could have supported our findings more robustly.

Another limitation of our study relates to the possible seasonality effects on the studied association, since the exposed and unexposed groups were recruited in different time frames, due to the COVID-19 exposure window. Few studies have shown variability in the risk of developing depression in different months of the year. While some studies showed that depression occurrence is usually high in spring season, others have failed to find such association [35,36,37].

Finally, our study group was evaluated 2 to 3 months following the diagnosis of the first confirmed case of COVID-19 in Israel. Since we did not include preterm deliveries, the exposure was mostly confined to the third trimester of pregnancy. This might not be sufficient time to determine the association between exposure to the COVID-19 pandemic and local outbreak and maternal cognitive function.

Future studies may be able to identify an accumulative effect of this exposure, as well as evaluating more accurately not only direct and immediate effects, but also effects that are related to more consequential life events and implications. More specifically, additional research may help achieve better understanding of immediate effects, as well as reveal long-term effects of this exposure on maternal health (e.g., recovery from postpartum depression or resolution of cognitive decline), and offspring physical and mental development.

The strengths of our study include use of reliable and validated tools for the assessment of objective and subjective cognitive performances with clear and strong results, which are consistent with previously published data. Our study also controlled for many possible confounding factors while comparing two very similar groups of postpartum women who delivered at the SUMC, with similar demographic, medical, obstetrical and psychiatric characteristics and history, including a direct assessment of maternal depression risk.

To the best of our knowledge, this is the first study to directly evaluate maternal subjective and objective cognitive function during the immediate postpartum period amid a large-scale crisis and natural disaster such as the COVID-19 pandemic. The association observed between the COVID-19 pandemic and cognitive assessments sheds additional light on the issue surrounding mental health of postpartum women during the COVID-19 pandemic, thereby allowing us to speculate regarding possible mechanisms affecting maternal cognitive abilities and well-being.

## 5. Conclusions

The COVID-19 pandemic does not appear to influence maternal objective cognitive function, yet our data suggest a significant decline in maternal subjective cognitive function during the pandemic. Further research is required in order to confirm our findings and determine that indeed the COVID-19 pandemic is not associated with a decrease in cognitive function in postpartum women, and to investigate other implications the pandemic may have on maternal and offspring mental health and general well-being.

## Figures and Tables

**Table 1 jcm-09-03727-t001:** Maternal clinical and demographic features.

Characteristics		During COVID-19 Pandemic *n* = 79 *n* (%)	Before COVID-19 Pandemic *n* = 123 *n* (%)	*p*-Value
Maternal age, years (mean ± SD)		28.1 ± 5.6	28.3 ± 5.1	0.836
Country of birth	Israel	85 (93.4)	105 (88.2)	0.206
	Other	6 (6.6)	14 (11.8)	
Familial status	Married	88 (98.9)	93 (97.9)	0.599
	Other	1 (1.1)	2 (2.1)	
Ethnicity	Jewish	40 (44)	67 (56.8)	0.066
	Bedouin	51 (56)	51 (43.2)	
History of non-psychiatric chronic illness		12 (14.1)	16 (13.4)	0.891
EPDS score ≥ 10		26 (28.6)	40 (34.5)	0.365
Gravidity	1	17 (20)	27 (22.7	0.798
	2–4	49 (57.6)	63 (52.9)	
	5+	19 (22.4)	29 (24.4)	
Time from admission to assessment (days)		1.0 ± 0.8	1.6 ± 1.2	<0.001

SD, standard deviation. EPDS, Edinburgh Postnatal Depression Scale.

**Table 2 jcm-09-03727-t002:** Pregnancy and delivery data.

Characteristics		During COVID-19 Pandemic *n* = 79 *n* (%)	Before COVID-19 Pandemic *n* = 123 *n* (%)	*p*-Value
Fertility treatments		3 (3.6)	10 (9.7)	0.101
Abnormal pregnancy screening tests ^1^		2 (2.2)	9 (7.6)	0.084
Pregnancy complications ^2^	GDM	3 (3.5)	5 (5)	0.634
	Preeclampsia	0 (0)	2 (1.7)	0.222
Birth weight (g)		3179.5 ± 407.5	3233.4 ± 410.6	0.355
Low birth weight		2 (2.4)	6 (5)	0.329
Gestational age at birth (weeks)		39.2 ± 1.1	39.3 ± 1.1	0.355

^1^ Abnormal pregnancy screening tests—including nuchal translucency, serum biomarkers, alpha-fetoprotein test, early and late ultrasound anatomical survey. ^2^ Pregnancy complications—including gestational diabetes mellitus (GDM) or preeclampsia.

**Table 3 jcm-09-03727-t003:** Objective and subjective cognitive tests results of women delivering during and before the COVID-19 pandemic.

Questionnaire	During COVID-19 Pandemic *n* = 79	Before COVID-19 Pandemic *n* = 123	*p*-Value
Objective cognitive test SDMT 90 (Mean ± SD)	44.1 ± 13.7	45.0 ± 11.9	0.660
Objective cognitive test SDMT4 (Mean ± SD)	116.9 ± 34.2	115.7 ± 30.5	0.794
Subjective cognitive test AFI (Mean ± SD)	70.0 ± 15.4	75.1 ± 14.7	0.018

SD, standard deviation. SDMT90, Symbol Digit Modalities Test for 90 s. SDMT4, Symbol Digit Modalities Test for 4 min. AFI, Attention Function Index.

**Table 4 jcm-09-03727-t004:** Multivariable regression model for the association between delivery during COVID-19 pandemic and maternal subjective cognitive test (AFI).

Variables	Beta	95% CI	*p*-Value
Delivery during COVID-19 pandemic (vs. delivery before the COVID-19 pandemic)	−5.608	−10.09; −1.13	0.014
Maternal age	0.279	−0.15; 0.7	0.198
Ethnicity	−0.854	−5.57; 3.86	0.721
Time from admission to assessment	−0.974	−3.15; 1.2	0.377

AFI, Attention Function Index.

**Table 5 jcm-09-03727-t005:** Multivariable regression model for the association between delivery during COVID-19 pandemic and maternal objective cognitive test (SDMT90).

Variables	Beta	95% CI	*p*-Value
Delivery during COVID-19 pandemic (vs. delivery before the COVID-19 pandemic)	0.662	−3.18; 4.51	0.734
Maternal age	−0.107	−0.47; 0.26	0.567
Ethnicity	−9.531	−13.56; −5.51	<0.001
Time from admission to assessment	0.363	−1.56; 2.28	0.710

SDMT90, Symbol Digit Modalities Test for 90 s.

**Table 6 jcm-09-03727-t006:** Multivariable regression model for the association between delivery during COVID-19 pandemic and maternal objective cognitive test (SDMT4).

Variables	Beta	95% CI	*p*-Value
Delivery during COVID-19 pandemic (vs. delivery before the COVID-19 pandemic)	1.277	−8.01; 10.56	0.786
Maternal age	−0.852	−1.73; 0.03	0.058
Ethnicity	−24.283	−34.12; −14.45	<0.001
Time from admission to assessment	−2.828	−7.56; 1.9	0.240

SDMT4, Symbol Digit Modalities Test for 4 min.

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
