# Peer review of "The Association between Delivery during the COVID-19 Pandemic and Immediate Postpartum Maternal Cognitive Function"

_jcm, 2020, doi:10.3390/jcm9113727_

Round 1

Reviewer 1 Report

1. This is an interesting paper about the possible influence of COVID-19 Pandemic to
the Postpartum Maternal Cognitive Function.
2. Considering that the Symbol Digit Modalities Test (SDMT) is a screening test
commonly used in clinical settings to assess neurological dysfunction (as in multiple
sclerosis), please add a “convincing” paragraph in the Introduction that this test could
be proved valuable in a broad spectrum of situations (including the Immediate
Postpartum Maternal Cognitive Function).
3. In the Abstract of your article, you use a general statement (lines 19 to 20): “Survivors
of previous epidemic periods were found to have higher rates of adverse mental
conditions”. However, in the Introduction (lines 48 to 50), you use a single reference
about 2003 SARS epidemic. Please, “equalize” these two sentences.

Author Response

  1. This is an interesting paper about the possible influence of COVID-19 Pandemic to the Postpartum Maternal Cognitive Function.

Response: We thank the reviewer for the comment.

  1. Considering that the Symbol Digit Modalities Test (SDMT) is a screening test commonly used in clinical settings to assess neurological dysfunction (as in multiple sclerosis), please add a “convincing” paragraph in the Introduction that this test could be proved valuable in a broad spectrum of situations (including the Immediate Postpartum Maternal Cognitive Function).
    Response: According to the comment of the reviewer, the following paragraph has been added to the Introduction section: “The SDMT is an accepted tool, implemented in a wide range of research areas and clinical settings, for the assessment of patients of different ages and conditions; including children and adults with multiple sclerosis, elderly patients with Alzheimer’s disease, and breast cancer patients [20-22]."

References:

  1. Kalb, R.; DeLuca, J. Recommendations for cognitive screening and management in multiple sclerosis care. Mult Scler 2018, 24, 1665-1680. [PubMed]
  2. Hoffmann, K.; Hasselbalch, S.G. Moderate-to-High Intensity Physical Exercise in Patients with Alzheimer's Disease: A Randomized Controlled Trial. J Alzheimers Dis 2016, 50, 443-453. [PubMed]
  3. Tong, T.; Cheng, Z. Efficacy of Acupuncture Therapy for Chemotherapy-Related Cognitive Impairment in Breast Cancer Patients. Med Sci Monit 2018, 24, 2919-2927. [PubMed]

(Introduction section, lines 122-124 in the revised manuscript)

  1. In the Abstract of your article, you use a general statement (lines 19 to 20): “Survivors of previous epidemic periods were found to have higher rates of adverse mental conditions”. However, in the Introduction (lines 48 to 50), you use a single reference about 2003 SARS epidemic. Please, “equalize” these two sentences.

Response: According to the comment of the reviewer, the words “previous epidemic periods” in the Abstract section, were replaced by the words “the 2003 SARS epidemic”.

(Abstract, line 19 in the revised manuscript)

Reviewer 2 Report

Postpartum Maternal Cognitive Function”, authored by Hamam et al, compared cognitive function in immediate postpartum period of women delivering during (22 women) and before (27 women) the COVID19 pandemic who delivered at the same medical center in Israel. Specifically, two groups scored similarly for an objective neurocognitive test Symbol Digit Modalities (SDM) test, whereas women delivering during the COVID-19 period had lower scores for a subjective self-estimation questionnaire Attention Function Index (AFI). The findings indicate that exposure to the COVID-19 pandemic is associated with a decrease in subjective, but not objective, cognitive function, during the immediate postpartum period. The authors speculated that such effect of maternal emotional state on cognitive function might be subjective. One concern is that only one objective assessment and one subjective assessment were included. Some other subjective and objective assessments or behavioral evaluation should be included to support this speculation.

The authors have taken into account a few pitfalls in this manuscript and discussed these issues, such as the study assessed mother’s cognition at a single time-point and subjects of two groups were recruited in different time frames. It’s important to discuss these pitfalls, as depression occurrence is usually high in spring season when COVID-19 group subjects were recruited.

I also have other below suggestions.

  1. Line 43. The word “spread” was misused in this sentence, as some original cases have been identified and reported in North America and Europe.
  2. Background of postpartum depression progress is needed in the Introduction, such as the duration of the development and progression. The delivery was around May, three months into the pandemic. Is the last trimester have greatest impact on development of postpartum depression?
  3. It would be important to learn long-term effects vs. immediate effects, how these two groups of women recover from postpartum depression. It’s also important to evaluate physical growth and social/mental development of infants.

Author Response

  1. One concern is that only one objective assessment and one subjective assessment were included. Some other subjective and objective assessments or behavioral evaluation should be included to support this speculation.
    Response: In our study we have implemented validated and well-established objective and subjective tests. Indeed, as only one objective assessment and one subjective assessment were included, other subjective and objective assessments or behavioral evaluation could have strengthened our speculation. Nevertheless, according the approval we have received from the local Institutional Review Board (IRB), no additional tests can be added to this study at this time. According to the comment of the reviewer, the following sentence has been added to the Discussion section: “The SDMT and the AFI constitute one objective and one subjective assessment tools respectfully, while the use of more subjective and objective tools could have supported our findings more robustly.”

(Discussion section, lines 289-291 in the revised manuscript)

  1. The authors have taken into account a few pitfalls in this manuscript and discussed these issues, such as the study assessed mother’s cognition at a single time-point and subjects of two groups were recruited in different time frames. It’s important to discuss these pitfalls, as depression occurrence is usually high in spring season when COVID-19 group subjects were recruited.
    Response: According to the comment of the reviewer, the following sentence has been added to the Discussion section: “Few studies have shown variability in the risk of developing depression in different months of the year. While some studies showed that depression occurrence is usually high in spring season, others have failed to find such association [35-37].”

References:

  1. Sylven, S.M.; Skalkidou, A. Seasonality patterns in postpartum depression. Am J Obstet Gynecol 2011, 204, 413. [PubMed]
  2. Overland, S; Colman, I. Seasonality and symptoms of depression: A systematic review of the literature. Epidemiol Psychiatr Sci 2019, 29, 31. [PubMed]
  3. Panthangi, V.; Reickert, E. Is Seasonal Variation Another Risk Factor for Postpartum Depression? J Am Board Fam Med 2009, 22, 492-497. [PubMed]

(Discussion section, lines 294-296 in the revised manuscript)

  1. Line 43. The word “spread” was misused in this sentence, as some original cases have been identified and reported in North America and Europe.
    Response: According to the comment of the reviewer, the words “spread globally” in the Introduction section, have been replaced by the words “become a global concern”.

(Introduction section, line 43 in the revised manuscript)

  1. Background of postpartum depression progress is needed in the Introduction, such as the duration of the development and progression.
    Response: According to the comment of the reviewer, the following sentences have been added to the Introduction section: “Postpartum depression occurs in up to 19% of all pregnancies, while the first six months after delivery are thought to bear increased risk for the onset of depression [11]. As per DSM-V definition, this condition occurs up to 4 weeks after delivery, yet many experts in the field define postpartum depression as occurring anytime within the first year postpartum, irrespective of the time of onset [12].”

References:

  1. O’hara, M.W.; McCabe, J.E. Postpartum depression: current status and future directions. Annu Rev Clin Psychol 2013, 9, 379-407. [PubMed]
  2. Parrigon, K.S.; Stuart, S. Perinatal depression: an update and overview. Curr Psychiatry Rep 2014, 16, 468. [PubMed]

(Introduction section, lines 67-71 in the revised manuscript)

  1. The delivery was around May, three months into the pandemic. Is the last trimester have greatest impact on development of postpartum depression?
    Response: Truly, the last trimester have greatest impact on development of postpartum depression According to the comment of the reviewer, the following sentence has been added to the Introduction section: “During pregnancy, the third trimester is when women face the greatest risk of developing depression [13].”

Reference:

  1. Okagbue, H.I.; Adamu, P.I.; Bishop, S.A.; Oguntunde, P.E.; Opanuga, A.A.; Akhmetshin, E.M. Systematic Review of Prevalence of Antepartum Depression during the Trimesters of Pregnancy. Open Access Maced J Med Sci 2019, 7, 1555-1560. [PubMed]

(Introduction section, lines 71-72 in the revised manuscript)

  1. It would be important to learn long-term effects vs. immediate effects, how these two groups of women recover from postpartum depression. It’s also important to evaluate physical growth and social/mental development of infants.
    Response: Indeed, it would be interesting to evaluate long-term influence of this exposure, how these two groups of women recover from postpartum depression and to evaluate physical growth and social/mental development of infants. We intend to do so in future studies. However, this is a single point study, and unfortunately the limitations of the current IRB approval do not allow us to evaluate long-term outcomes at this time. According to the comment of the reviewer, the following sentence has been added to the Discussion section: “More specifically, additional research may help achieve better understanding of immediate effects, as well as reveal long-term effects of this exposure on maternal health (e.g. recovery from postpartum depression or resolution of cognitive decline), and offspring physical and mental development.”

(Discussion section, lines 304-307 in the revised manuscript)
